# A Review of the Health Protective Effects of Phenolic Acids against a Range of Severe Pathologic Conditions (Including Coronavirus-Based Infections)

**DOI:** 10.3390/molecules26175405

**Published:** 2021-09-06

**Authors:** Sotirios Kiokias, Vassiliki Oreopoulou

**Affiliations:** 1European Research Executive Agency, Place Charles Rogier 16, 1210 Bruxelles, Belgium; 2Laboratory of Food Chemistry and Technology, School of Chemical Engineering, National Technical University of Athens, Iron Politechniou 9, 15780 Athens, Greece; vasor@chemeng.ntua.gr

**Keywords:** phenolic acids, antioxidants, health properties

## Abstract

Phenolic acids comprise a class of phytochemical compounds that can be extracted from various plant sources and are well known for their antioxidant and anti-inflammatory properties. A few of the most common naturally occurring phenolic acids (i.e., caffeic, carnosic, ferulic, gallic, p-coumaric, rosmarinic, vanillic) have been identified as ingredients of edible botanicals (thyme, oregano, rosemary, sage, mint, etc.). Over the last decade, clinical research has focused on a number of in vitro (in human cells) and in vivo (animal) studies aimed at exploring the health protective effects of phenolic acids against the most severe human diseases. In this review paper, the authors first report on the main structural features of phenolic acids, their most important natural sources and their extraction techniques. Subsequently, the main target of this analysis is to provide an overview of the most recent clinical studies on phenolic acids that investigate their health effects against a range of severe pathologic conditions (e.g., cancer, cardiovascular diseases, hepatotoxicity, neurotoxicity, and viral infections—including coronaviruses-based ones).

## 1. Introduction

Free radicals are mainly reactive oxygen species (ROS) (including hydroxyl-, superoxide-radicals and singlet oxygen) that are formed in tissue cells by various endogenous and exogenous pathways. ROS normally exert an adverse impact on human health by inducing the so called “oxidative stress conditions” [1]. The ability of free radicals to structurally modify cellular components and cause oxidative damage to biomolecules (LDL-low density lipoproteins, DNA, etc.) has revealed their involvement in a variety of health pathologies (i.e., inflammation, aging, types of cancer and cardiovascular diseases) [2,3].

Nature has generously offered several types of natural dietary antioxidants, among which phenolic compounds can operate as scavengers of free radicals in vivo and can efficiently reduce the harmful health impacts of oxidative damage [4,5]. Phenolic acids comprise a group of natural phenolic compounds that are present in a wide range of herbs and other species of the plant kingdom [6]. More specifically, thyme, oregano, rosemary, sage, and mint herbal preparations—all rich in various phenolics—have been reported to exert strong antioxidant biochemical and anti-inflammatory properties [7,8]. A few authors have reviewed the radical scavenging capacity of phenolic acids and their subsequent beneficial effects against the development of cancer, cardiovascular diseases and other health disorders (such as skin problems, inflammations, bacterial infections, etc.) [9]. The main biochemical pathways and mechanisms of phenolic actions against the development of certain types of cancer include: free radical scavenging, enzyme induction, DNA damage repair, cell proliferation depression, and apoptosis [10].

In their recent publication, Kiokias et al. (2020) [11] focused on the in vitro antioxidant activities of a few common naturally occurring phenolic acids (caffeic, carnosic, ferulic, gallic, p-coumaric, rosmarinic, vanillic) against the oxidation of oil-in-water emulsions. Such interfacial lipid-based systems generally mimic the structure of biological membranes and biomolecules that when attacked by free radicals are prone to harmful in vivo oxidative reactions.

In this paper, the authors first report on the main structural features of phenolic acids as well as on a few important natural sources and their extraction techniques. Subsequently, the main focus of this analysis is to provide an overview of the most recent clinical studies on phenolic acids that investigate their health effects against a few severe pathologic disorders.

## 2. Structure, Herbal Sources and Extraction of the Most Common Naturally Occurring Phenolic Acids

### 2.1. Structural Classidication of Natural Phenolic Acids

In terms of their chemical structure, phenolic acids are classified as:

Hydroxybenzoic acids with a C_6_-C_1_ structure: Among them a trihydroxy derivative (gallic acid) has been associated with tea antioxidant activity, while vanilic acid is a methoxy-hydroxy derivative serving as a well-known flavouring agent [12].

Hydroxycinnamic acids with a C_6_-C_3_ structure [13]. These are abundant in plant sources, with p-coumaric (4-hydroxy derivative), caffeic (3, 4-dihydroxy derivative) and ferulic (3-methoxy, 4-hydroxy derivative) commonly present in various culinary herbs. In addition, rosmarinic acid (an ester of caffeic acid with 3,4-dihydroxyphenyl lactic acid) is mainly encountered in certain aromatic herbs [14].

Phenylacetic acids with a C_6_-C_2_ structure. Phenylacetic acids are scarce in fruits and vegetables, while a dihydroxy derivative was detected in strawberry tree honey [15]. Carnosic acid belongs to the phenolic diterpenes that are usually classified as hybrid phenolics [13].

This review focuses on the most common hydroxybenzoic and hydroxycinnamic phenolic acids, along with carnosic acid, the chemical structures of which are given in Figure 1.

### 2.2. Herbal Sources and Extraction of Phenolic Acids

Caffeic acid (CA) is found at high levels in various herbs worldwide, including the South American herb *yerba mate* (1.5 g/kg) [16], the Japanese herbal leaf tea, the tea stem from *Moringa oleifera* L. [17], and thyme (1.7 mg/kg) [18].

Carnocic acid (CarA) can be found in a few species of the Lamiaceae family (such as rosemary and common salvia species). It has been reported to be present at a concentration of 1.5 to 2.5% in dried sage leaves [19,20].

Ferulic acid (FA) is present in black beans at an average concentration of 0.8 g/kg, while flaxseed has been reported as the richest natural source of FA glucoside (4.1 ± 0.2 g/kg), [21,22]. FA has been also identified as the major phenolic acid in *Angelica sinensis* (Oliv.), a traditional medicinal and edible plant in China [23].

Gallic acid (GA) has been found to be the main phenolic acid in tea [24] but also reported in high amounts in the parasitic plant *Cynomorium coccineum*, the aquatic plant *Myriophyllum spicatum*, and the blue-green alga *Microcystis aeruginosa* [25]. In addition, GA was recently identified as the main phenolic compound in leaf extracts from the medicinal halophyte *Thespesia populnea* tea [26].

p-Coumaric acid (p-CA) has been identified in basil, garlic [27] and in amaranth leaves and stem at a concentration range of 28–44 mg/kg [28]. p-CA has been reported as the major active compound in *Bambusae Caulis*, a Chinese medicinal herb [29] as well as in cultivars of husked oat *(Avena sativa* L.) in Finland [30].

Rosmarinic acid (RA) is the main phenolic component in several members of the Lamiaceae family, including among others: *Rosmarinus officinalis, Origanum* spp., *Perilla* spp., and *Salvia officinialis* in concentrations varying between 0.05 and 26 g/kg dry weight [31]. Additionally, the results of Tsimogiannis et al. [32] indicate an amount of 19.5 g/kg in the leaves of pink savoury (*Satureja thymbra* L.).

Vanillic Acid (VA) is commonly found in several fruits, olives, and cereal grains (e.g., whole wheat), as well as in wine. VA was also identified in fruit extracts of the açaí palm plant (*Euterpe oleracea*) [33] and in the root of *Angelica sinensis* (an herb indigenous to China) at concentrations between 1.1 and 1.3 g/kg [34].

### 2.3. Extraction of Phenolic Acids from Their Natural Sources

The extraction and identification of phenolic acids has been studied by various researchers [35,36]. Phenolic acids are compounds with medium to high polarity and, therefore, can be extracted by water [37]. Nevertheless, aqueous solutions of ethanol or acetone (50–70%) are the best solvents for the quantitative extraction of hydrocinnamic acids [38]. On the contrary, CarA exhibits low polarity and is quantitatively extracted with the use of pure acetone or ethanol [39].

Hydroxycinnamic and hydroxybenzoic acids may be linked to polysaccharides of the cell walls by ester bonds and to lignin components by ester or ether bonds [40]. Mild alkaline hydrolysis can be implied to cleave the ester bonds, while acid hydrolysis to cleave the ether bonds and release the phenolic acids [41]. However, phenolic acids may be degraded under alkaline conditions, e.g., RA has been reported to transform to CA [42]. Additionally, mild temperature and time combinations are suggested to avoid degradation. The most prone to degradation is CarA, which is oxidised to carnosol (which also exhibits antioxidant activity) at temperatures higher than 50 °C and at longer extraction times [43].

In addition to conventional solid liquid extraction, ultrasound assisted extraction and microwave assisted extraction proved even more effective for phenolic acid extraction, while shortening extraction time [44,45].

The predominant role of high-performance liquid chromatography (HPLC) in the definition of the phenolic profile of various plant sources has been recently examined by Ciulu et al. (2018) [46], who also present the most recently developed mass spectrometry-based detection systems. In addition, the various developed procedures for the quantification of phenolic compounds have been described in the literature, along with the spectrophotometric protocols for the evaluation of their antioxidant properties [47,48].

## 3. Biochemical and Health Properties of the Examined Phenolic Acids

The leading cause of a few severe human health disorders is oxidative stress, a consequence of overproduction and accumulation of free radicals [49]. Naturally occurring polyphenols have been shown to possess a number of biological activities such as antibacterial, antiviral, anticancer, and anti-cholesterol properties [50]. This section reports on the most recent in vitro (in human cell lines) and in vivo (clinical animal) trials on phenolic acids. The analysis focuses on the specific health diseases, i.e., cardiovascular diseases, cancer, hepatotoxicity, neurodegenerative disorders, and microbial or viral infections including COVID-19, that all together account for the majority of deaths in the western world. The main research findings about the clinical effects of phenolic acids per health disease are discussed in the following paragraphs while a summary of the most recent studies is presented in the overview Tables (Table 1 and Table 2).

### 3.1. Effects against Cancer

Globally, cancer is the second leading cause of death. In the continuous search for safer and more effective treatments than chemotherapy or radiotherapy, plant phenolic acids have gained importance displaying a great prospective as cytotoxic anti-cancer agents promoting apoptosis, reducing proliferation, and targeting various aspects of cancer [51].

#### 3.1.1. Individual Phenolic Acids

The chemical structure of CA and mainly the presence of free phenolic hydroxyls is believed to strongly account for its antioxidant capacities that, in turn, link to certain anti-carcinogenic properties [52]. According to the literature, though, CA phenyl ester (CAPE) is actually the natural CA derivative with the most dominant anticancer activities. Wang et al. [53] reported a significantly enhanced suppression of tumour growth in mice treated with CAPE based nanoparticles, revealing thereby its potential use in anticancer nanomedicine.

Zhang et al. [54] observed that FA significantly decreased tumour volume and increased apoptosis in an MDA-MB-231 xenograft mouse model, thereby acting as an effective therapeutic agent against breast cancer. More recently, Al-Mutairi et al. [55] investigated the combination effect of lower doses of thymoquinone and FA on the proliferation, apoptosis, and cell cycle of breast cancer cell line MDA-MB-231. The authors reported that either 25 µM of thymoquinone or 250 µM of FA, individually, had no effect but in combination significantly reduced cell proliferation, thus exerting an anticancer therapeutic potential. Solomonov et al. [56] demonstrated a significant anti-inflammatory effect of CarA combined with astaxanthin and a lycopene-rich tomato extract in a nutrient supplementation.

Sung and Wang [57] treated human cells (EC9706 and KYSE450) with different GA concentrations (10, 20, and 40 μg/mL). According to the results, GA decreased the growth of xenograft tumour in vivo and promoted cell apoptosis in a concentration-dependent manner. Sales et al. [58] conducted a study to evaluate the effect of GA isolated from methanolic fruit extract of *Terminalia bellirica* to inhibit the survival of breast cancer cells (MCF-7 & MDA-MB-231). The authors reported that GA at 80 μM exhibited decreased the survival of cancer cells and induced apoptosis, revealing its potential as an anticancer agent to be further explored for breast cancer drugs.

The anticancer and antitumoral properties of RA have been reviewed by Afonso et al. [59]. During 2020, two new clinical animal studies were conducted on RA. Luo et al. [60] reported that RA inhibited the proliferation and negatively affected the migratory potential of human oral cancer cells (cell line SCC-15) with a dose-dependent effect. Messeha et al. [61] reported that RA caused significant cytotoxic and antiproliferative effects in two racially different triple-negative breast cancer (TNBC) cell lines in a dose- and time-dependent manner.

Anbalagan et al. [62] explored the antioxidant efficacy of VA in dimethylbenz[a]anthracene (DMBA)-induced oral carcinogenesis. Supplementation with VA (200 mg/kg body weight) for 14 weeks significantly restored the disturbances in antioxidant status {superoxide dismutase, catalase) to near normal range in DMBA treated hamsters.

Furthermore, in vivo experiments confirmed that treatment with VA caused significant inhibition of tumour growth in a xenografted tumour model [63]. The in vitro antioxidant capacity of VA was demonstrated through the reduced DNA damage, induced by H_2_O_2_ in human lymphocytes at concentrations of 0.17–67.2 μg/mL [64]. These studies reveal that VA provides new perspectives of phenolic antitumour activity.

#### 3.1.2. Natural Extracts Rich in Phenolic Acids

A body of research has been conducted in recent years about the anticancer properties of herbal extracts rich in various phenolic acids. Jeong et al. [65] observed clear therapeutic effects of polyphenolic mixtures (containing among others GA, p-CA and ellagic acid) against lung cancer cells.

Hydroxycinnamic acid derivatives of mulberry fruits were reported to increase the production of reactive oxygen species by acting as pro-oxidants and hence killing the cancer cells [66]. In another study, p-CA rich methanolic extracts of *Amaranthus spinosus* and of *Amaranthus caudatus* L. were shown to possess significant anti-inflammatory activity in mouse models [67]. Fernadez et al. [68] have reported that bael (*Aegle marmelos*) flowers and tulsi (*Ocimum tenuiflorum*) seeds (rich in GA, p-CA, CA and VA) present a strong antioxidant character against DNA damage.

Koyuncu [69] conducted a study on human colon (DLD-1), endometrium (ECC-1) cancer cells and embryonic kidney (HEK-293) cells to examine the anti-cancer and antioxidant properties of the methanolic extract obtained from *Artemisia absinthium* L. species. According to the results, the *A. absinthium* extract, rich in various phenolic acids, showed an antioxidant effect and a cytotoxic activity on DLD-1 and ECC-1 cancer cells. Waheed et al. [70] noted that phenolic acids (CA and GA) are the most important ingredients of honey with known anti-cancer activity and their main suggested mechanisms are antioxidant, apoptotic, tumour necrosis, anti-inflammatory and estrogenic effects.

### 3.2. Effects against Cardiovascular Diseases

In spite of the medical advances, cardiovascular diseases (CVDs) remain a significant concern, causing the highest number of mortality cases globally and imposing a great burden upon the economies and public health of nations. Research involving both animal and human cells has proven that mixtures of phenolic acids possess cardioprotective properties such as anti-hypertensive, anti-hyperlipidemia and anti-hypertrophy activity [71].

#### 3.2.1. Individual Phenolic Acids

Silva and Lopez [72] reviewed the cardiovascular effect of CA and its derivatives. The authors claimed that their antioxidant, anti-inflammatory and anti-angiogenic properties contribute to an important anti-atherosclerotic effect and protect tissues against ischemia/reperfusion injuries and the cellular dysfunction caused by different physico-chemical agents. Besides, Olas et al. [73] highlighted the antioxidant and antiplatelet potential of CA, the dietary supplementation of which may ameliorate CVDs through various mechanisms, such as by decreasing oxidative stress and inhibiting blood platelet activation.

Salazar-López et al. [74] supplemented male Wistar rats with either lard at 310 g/kg (HFD) or lard and FA at 2 g/kg (HFD + FA) for eight weeks. The rats fed with HFD + FA had significantly lower plasma lipids and glucose levels compared with the HFD group. Bumrungpert et al. [75] conducted a randomized, double-blind, placebo-controlled trial with hyperlipidemia rats. The treatment group (n = 24) was given FA (1000 mg daily) and the control group (n = 24) was provided with a placebo for six weeks. FA supplementation demonstrated a statistically significant decrease in total cholesterol (8.1%; *p* = 0.001), LDL-C (9.3%; *p* < 0.001), and triglyceride (12.1%; *p* = 0.049) and increased HDL-C (4.3%; *p* = 0.045) compared with the placebo, while oxidized LDL-C was significantly decreased in the FA group (7.1%; *p* = 0.002). The results of both studies revealed FA’s potential to reduce cardiovascular disease risk factors.

Akbari [76] noted that GA exerts a protective action against CVDs by increasing antioxidant enzyme capacity and inhibition of lipid peroxidation and decreasing serum levels of cardiac marker enzymes.

A study by Ibitoye and Ajiboye [77] investigated the influence of CA, FA and GA on high-fructose diet-induced metabolic syndrome in rats. The authors reported that oral administration of the phenolic acids significantly reversed the increase in the levels of lipid parameters and indices of atherosclerosis, cardiac and cardiovascular diseases.

Sherratt et al. [78] noted that RA and its esters inhibit membrane cholesterol domain formation through an antioxidant mechanism based on alkyl chain length.

#### 3.2.2. Natural Extracts Rich in Phenolic Acids

Murino Rafacho et al. [79] reported that daily dietary supplementation (11–110 mg) of rosemary leaves (particularly rich in RA) attenuated cardiac remodelling on male Wistar rats by improving energy metabolism and decreasing oxidative stress. The findings support further investigations of the use of rosemary as an adjuvant therapy against myocardial infarction.

In a study by Faponle et al. [80], 48 adult male rats were supplemented with a leaf extract of *Amaranthus spinosus* (particularly rich in p-CA) at a single dose of 250 mg/kg continuously for 28 days. Although no significant alterations were observed in the cholesterol and triglyceride levels of the heart, there was a significant decrease in the atherogenic indices of plasma, revealing a potential protective role against CVD related disorders. Very recently, Fatma et al. [81] reported a clear protective effect of *Thymus algeriensis* against hydrogen peroxide induced cardiotoxicity in rats.

Panda et al. [82] investigated the cardioprotective activity of the *Macrotyloma uniflorum* seed extract (MUSE) and its phenolic acids (p-CA and FA) in isoproterenol (ISO)-induced myocardial infarction in rats. Treatment of rats with MUSE (250 and 500 mg/kg) for 30 days resulted in a significant attenuation of serum marker enzymes, total cholesterol, triglycerides, uric acid and restoration of heart rate, systolic, diastolic and mean arterial pressure.

Cianciosi et al. [83] reviewed the phenolic compounds in honey and their associated health benefits. The authors reported that the abundance of phenolic acids in honey (CA, FA, VA, etc.) may account for its protective effect in the cardiovascular system where it mainly prevents the oxidation of low-density lipoproteins.

### 3.3. Effects against Hepatotoxicity and Liver Disorders

The liver plays a crucial role in the regulation of various physiological processes and in the excretion of endogenous waste metabolites and xenobiotics. The plant kingdom is full of liver protective chemicals such as phenols, carotenoids, flavonoids, and phenolic acids [84].

#### 3.3.1. Individual Phenolic Acids

Ajiboye et al. [85] conducted a study to evaluate the influence of CA on 1,3-dichloro-2-propanol-induced hepatotoxicity in rats that received distilled water or CA (10 or 20 mg/kg body weight) for seven days. The authors reported that CA protects against 1,3-dichloro-2-propanol-induced hepatotoxicity by enhancing the cytoprotective enzymes and lowering inflammation. In addition, Mu et al. [86] observed that CA protects transplanted livers from injury, which is likely attributed to its protection of oxidative damage by interfering in PDIA3-dependent activation of NADPH oxidase.

Very recently, Hao et al. [87] explored the potential protective effect of CAPE on the cadmium-induced liver damage of 40 male mice that were treated daily with 10 μmol CAPE/kg body weight, gavage for four weeks. The authors concluded that CAPE administration significantly reduced cadmium level and improved liver tissue histopathology reporting for the first time a CAPE’s protection against CdCl_2_-induced hepatotoxicity.

Chen et al. [88] conducted a dietary supplementation of fish (*Megalobrama amblycephala*) with FA at 50–100 mg/kg body weight doses. The authors reported that FA significantly decreased the contents of pro-inflammatory cytokines such as TNF-α and IL-1β, thereby proving that this phenolic acid alleviates lipopolysaccharide-induced acute liver injury in fish.

Owumi et al. [89] studied the effects of GA against hepatoxicity in rats exposed to aflatoxin B_1_AFB_1_ (75 µg/kg body weight) and treated with GA (20 or 40 mg/kg) for 28 successive days. The authors concluded that GA ameliorated AFB_1_-induced hepatorenal dysfunction by decreasing oxidative stress and inflammation in rats.

Hussein et al. [90] evaluated the effects of GA and FA against an experimentally induced liver fibrosis by thioacetamide (TAA). Supplementation of rats with both FA and GA at 20 mg/kg/day, for six weeks exhibited hepatoprotective and antioxidant effects against TAA-induced liver fibrosis (mediated through inhibition of TGF-β1/Smad3 signalling and differentially regulating the hepatic expression level of miR-21, miR-30 and miR-200).

Lee et al. [91] reported that CarA modulates increased hepatic lipogenesis and adipocyte differentiation in ovariectomized mice fed with normal or high-fat diets.

Oguz et al. [92] exposed 32 rats to hepatic ischaemia/reperfusion (I/R) injury and subsequent treatment with an RA dose of 50 mg/kg via oral gavage. According to the results, RA significantly reduced liver function test parameters and decreased oxidative stress and abnormal histopathological findings in the liver.

#### 3.3.2. Natural Extracts Rich in Phenolic Acids

A few authors have very recently examined the effect of phenolic rich extracts against carbon tetrachloride (CC14)-induced liver injury in mice. Meng et al. [93] have concluded a clear inhibitory activity of *S. officinalis* rich in GA (8 mg/g). Meharie et al. [94] also reported a similar beneficial effect of *Clutia abyssinica* (Euphorbiaceae) against mice hepatotoxicity.

In an in vitro study, Hewage et al. [95] evaluated the cytotoxicity and hepatoprotective effect of different solvent fractions (aqua, butanol, chloroform, ethyl acetate and hexane) of *S. quelpaertensis Nakai* leaf. Between the five fractions (0–1000 µg/mL) only the ethyl acetate fraction, rich in phenolic acids (such as p-CA) showed a hepatoprotective effect against HepG2 cells.

Furthermore, Mbarki et al. [96] reported a clear protective effect of *Trigonella foenum graecum* (Fenugreek seeds), an extract rich in various phenolic acids, against CC14-induced damage in liver and kidney of male rats.

### 3.4. Effects against Neurological Disorders

#### 3.4.1. Individual Phenolic Acids

Alzheimer’s disease (AD) is an ultimately fatal brain disorder, which along with other chronic neurodegenerative conditions has imposed an increasingly large burden on social care systems [97]. Various medicinal plants rich in phenolic compounds were reported to exert a beneficial effect in the treatment of AD.

Habtemariam et al. [98] related the anti-AD therapeutic potential of CA with the presence of diorthohydroxyl (catecholic) aromatic moiety and also reviewed the neuroprotective effect of the two most common CA conjugated natural bioactive derivatives (chlorogenic acid and CAPE). Additionally, it was reported that CA improved behavioural impairments, and attenuated loss of dopaminergic neurons in mice, thereby exerting a clear neuroprotective effect [99]. Ferreira et al. [100] assessed the neurotrophic and neuroprotective effects of CAPE against cisplatin-induced neurotoxicity in PC12 cells. The authors reported that CAPE (10 μM) attenuated the inhibition of neuritogenesis and the downregulation of markers of neuroplasticity induced by cisplatin (5 μM). A recent medical study [101] reported clear inhibitory effects of CAPE against acetylcholinesterase, an enzyme linked with the development of AD.

Bahri et al. [102] noted that CarA can have a protective effect against chronic neurodegenerative conditions, like Parkinson’s disease, via a mechanism that links to the transcriptional activation of antioxidant Nrf2/ARE pathway.

Rehman et al. [103] observed a clear anti-inflammatory effect of FA against LPS-induced neuroinflammation in the mouse brain. Mori et al. [104] supplemented orally transgenic mice with epigallocatechin-3-gallate (EGCG) and/or FA (30 mg/kg each) daily for three months. The authors reported that the combined EGCG-FA treatment reversed cognitive impairment in most tests of learning and memory, presenting thereby an AD therapeutic effect.

Much attention has been given very recently to the potential beneficial effect of GA on mental health. Shabani et al. [105] reviewed several clinical studies and concluded that GA is effective against nervous system disorders, including AD, Parkinson’s disease, ischemia, depression and anxiety. In addition, Liu et al. [106] observed a clear neuroprotective effect of GA following the systemic administration of 100 mg/kg body weight to neuroinflammatory rat, compared to vehicle-treated rats. Khoshnam et al. [107] reported that GA (1.0 μM) protected against neurotoxicity in hippocampal neurons isolated and co-cultured with glial cells.

Rizk et al. [108] reported that oral administration of rats with ellagic acid (10 mg/kg/day) and RA (75 mg/kg/day) for 14 days mitigated the neural changes induced by doxorubicin, a chemotherapeutic agent. Very recently, Salau et al. [109] reported a clear neuroprotective activity of VA against Fe^2+^-induced oxidative toxicity in brain tissues (neuronal cell lines—HT22). Similarly, VA was found to exert clear neuroprotective effects and restore the spatial memory in rats, following VA supplementation for 14 consecutive days [110].

#### 3.4.2. Natural Extracts Rich in Phenolic Acids

El-Sawi et al. [111] examined the neuro-therapeutic properties of *Salvia splendens* plant cultivated in Egypt (particularly rich in RA and CA). The authors reported that dietary supplementation of rats with a dose of 500 mg/kg body weight of methanolic extracts of Salvia species for 4 weeks significantly attenuated their AlCl_3_-induced behavioural impairment (similar to that of AD).

During AD, the level of acetylcholine (AChE) in the brain is decreased whereas the level of oxidative reactive species increases and accumulation of β-amyloid protein starts. A black sesame pigment (extract of black sesame seeds rich in VA) upon simulation by gastrointestinal digestion was reported to have AChE activity [112]. In addition, an anti-amyloid aggregation activity of black sesame pigment was noted by Panzella et al. [113], a finding that may offer new perspectives towards its use as a food supplement for the prevention of AD. Liang et al. [114] highlighted the neuroprotective effect of *Fagopyrum dibotrys*, a natural extract rich in phenolic acids, against AD.

### 3.5. Protective Effects against Microbial and Viral Infections (Incl. COVID-19)

In addition to their antioxidant activities, plant-derived phenolic acids have been reported to exert antimicrobial and anti-inflammatory properties [115]. Over the past few years, herbal extracts and various essential oils rich in phenolics have also shown effective antifungal activities. More specifically, nanohydrogels embedded with natural plant extracts and oils have become the primary choice of pharmaceutical scientists [116]. In addition, a few authors have focused on various botanical sources with antimicrobial properties by exploring their classification, chemical composition and functional properties. Semeniuc et al. [117] examined a range of botanical essential oils (parsley, lovage, basil, thyme) using various chemometric methods and concluded that thyme essential oil exhibited the stronger antibacterial activity. An overview of the most recent studies on the potential activity of phenolic acids against microbial relevant infections is given in Table 2.

#### 3.5.1. Antimicrobial Activity of Phenolic Acids-Mechanism of Action

Liu et al. [118] noted that the chemical structure of phenolic acids enables their potential incorporation into biomaterial scaffolds, thus providing naturally derived functionalities that could improve healing outcomes. The chemical structure, position and number of substitutions in benzene rings seem to determine the anti-microbial activity of phenolic acids [119]. An increase in alkyl chain lengths enhances activity while hydroxybenzoic and hydroxycinnamic acids show different modes of action depending on the number of hydroxy and methoxy functional groups [120].

Wu et al. [121] reviewed the activity of various naturally derived phenolic acids with diverse skeletons and mechanisms, and concluded that CA and GA and their derivatives, especially, could provide us with an excellent source of novel antiviral drugs. Paulo and Santos [122] examined how incorporation of caffeic-ethyl cellulose microparticles in skin care products can offer antimicrobial and anti-aging protection. Moreover, Langland et al. [123] has reported an antiviral activity of CA towards herpes simplex (HSV), VSV-Ebola pseudotyped and vaccinia viruses. The authors reported that the antiviral activity increased and occurred early in the virus replication cycle with the addition of chelated inorganic ions or a metal such as iron to CA. Zhang et al. [124] observed that the cocktail of either CA (1.5 mg/mL) or chlorogenic acid (3 mg/mL) with the antibiotic fosfomycin (50 mg/L) was able to significantly inhibit the growth of the pathogen *Listeria monocytogenes*.

De Camargo et al. [125] reported that phenolic acid-rich extracts from peanut sources (24-49 μg phenolics/mL) exhibited a high antibacterial effect against the growth of Gram-positive (*Bacillus cereus*, *Staphylococcus aureus*, *Listeria monocytogenes*, etc.) and Gram-negative bacteria (*Pseudomonas aeruginosa*, *Salmonella Enteritidis*, *Escherichia coli*, etc.).

Baidoo et al. [126] investigated the wound healing, antimicrobial and anti-oxidant activities of methanol extracts of the leaf and stem of *E. africana* tested in the dermal excision wound model in rats. The methanol extract of bark stem, having GA as the main phenolic compound, demonstrated antibacterial activity against *Staphylococcus aureus* and *Streptococcus pyogenes* with a minimum inhibitory concentration of 1.56 mg/mL and thereby a great potential for the treatment of open wounds.

#### 3.5.2. Potential Protective Effects of Phenolic Acids against Coronavirus-Based Infections

Coronaviruses are the causative agents of many infectious diseases in humans and animals. These included severe acute respiratory syndrome (SARS), avian infectious bronchitis (IBV) in poultry, Middle East respiratory syndrome (MERS), and coronavirus disease 2019 (COVID-19) in humans [127]. The majority of publications on the herbal remedies of coronavirus, MERS, or SARS focused primarily on the use of polar compounds, including phenolic acids (namely CA) and flavonoids (namely quercetin and myricetin) [128].

Upon viral infection, heat shock protein A5 (HSPA5) is upregulated, then translocated to the cell membrane where it can recognize the SARS-CoV-2 spike. Elfiky et al. [129], using molecular docking and molecular dynamics, tested a few natural product compounds against the HSPA5 substrate-binding domain β (SBDβ). The results show a high to moderate binding affinity for a range of phenolic acids. More specifically, CA, CAPE, and p-CA may bind to cell-surface HSPA5, competing for its recognition by viral spike protein and contradicting its attachment. These compounds can be successful as anti-COVID-19 agents for people with a high risk of cell stress, like elders, cancer patients, and front-line medical staff. Similarly, Kumar et al. [130] used the strengths of molecular dynamics simulations and concluded that since the natural phenolic compounds are easily available/affordable, they may even offer a timely therapeutic/preventive value for the management of the SARS-CoV-2 pandemic.

Khalil and Tazeddinova [131] concluded that CA and GA could be considered in the treatment of COVID-19 and its related symptoms. Moreover, GA and CA were found to exert sustainable anti-viral activity against human coronavirus NL63 (HCoV-NL63), one of the main circulating coronaviruses worldwide that causes respiratory tract diseases like runny nose, cough, bronchiolitis and pneumonia [132].

Adem et al. [133] screened 27 CA derivatives against five proteins of SARS-CoV-2 aiming to evaluate the anti-viral efficacy of these natural bioactive entities against COVID-19 via molecular docking and molecular dynamics simulation. The obtained results have uncovered khainaoside C, 6-O-Caffeoylarbutin, khainaoside B, khainaoside C and vitexfolin A as potent modulators of COVID-19, possessing more binding energies than nelfinavir against COVID-19. Very recently, rosmarinic acid was claimed to have a demonstrated potential to increase the activity or expression of ACE-2 and could therefore aggravate SARS-CoV-2 effects [134].

## 4. Conclusions and Prospects

Based on this analysis of the in vitro and in vivo biochemical activities of phenolic acids, the authors have drawn the following conclusions:

A number of recent in vivo animal clinical trials and in vitro human cell studies offered sufficient evidence to support that the examined phenolic acids possess health protective effects against several pathogenic conditions, including cancer, cardiovascular, liver, and neurodegenerative diseases, as well as microbial infections. Based on their strong antiviral activities, phenolic acids could also be considered in the development of medical treatments against the spread of COVID-19 and its related symptoms.

Concerning the mechanism of actions we can note that: (a) the phenolic acid activity against types of cancer and cardiovascular diseases may respectively link to their established antioxidant effect against DNA damage and LDL oxidative deterioration; (b) the antimicrobial activity of phenolic acids is well determined by their structure and presence of functional groups (with CA and GA exerting the strongest reported activities)—although phenolic acids have shown clear neuro-protective effects, their exact mode of activity against neurotoxicity needs some further elucidation before further clinical developments; and (c) a number of botanical extracts rich in various phenolic acids have been increasingly shown to exert strong antioxidant and biochemical properties, a fact that may be associated with the synergistic effects of their individual phenolic compounds.

About future prospects in this scientific field, the authors would like to note that: (i) although the pharmacokinetic and non-toxic profile of phenolic acids make them suitable for clinical studies, in vivo human trials are needed to further explore their potential for extensive pharmaceutical applications; and (ii) since a number of recent model studies came out with promising results about the therapeutic potential of phenolic acid-based drugs for treatment or prevention of COVID-19, further in vitro and in vivo studies should be performed to clarify and evaluate the specific antiviral effects of these phytochemicals.

## Figures and Tables

**Figure 1 molecules-26-05405-f001:**
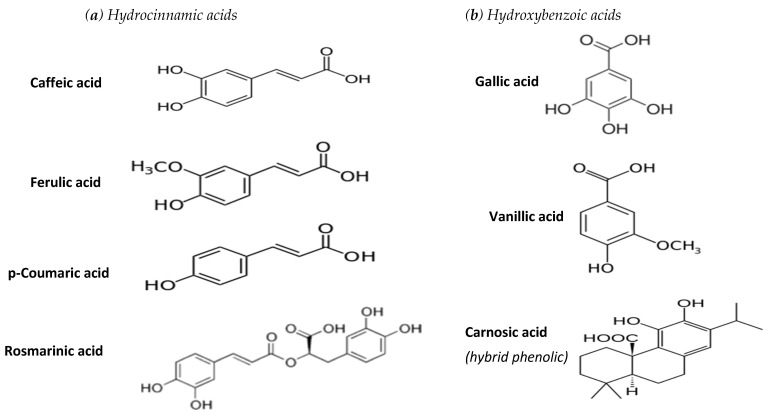
Chemical structure of phenolic acids examined in this study.

**Table 1 molecules-26-05405-t001:** Selection of in vivo and in vitro studies on the health/biochemical properties of various phenolic acids.

Health Disease	Phenolic Treatment & Conditions	Conclusion of Study/Health Effect	References
ANTICANCER PROTECTION	Effect of Thymoquinone (TQ-25 µM) and FA (250 µM) on proliferation and apoptosis of a breast cancer cell line MDA-MB 231.	FA in combination with TQ significantly reduced cell proliferation/anticancer effect	[55]
	Human EC cells (EC9706 and KYSE450) were treated with different concentrations (10–40 μg/mL) of GA	GA reduced the growth of xenograft tumour and promoted apoptosis in a concentration dependent manner.	[57]
	Rats subject to DMBA induced oral carcinogenesis were supplemented with VA (200 mg/kg bw p.o) for 14 weeks	VA significantly restored the disturbances in antioxidants status {superoxide dismutase, catalase) to near normal range in DMBA treated hamsters/anti-cancer effects	[62]
CARDIO-PROTECTION	Male Wistar rats supplemented with either lard at 310 g/kg (HFD) or lard and FA at 2 g/kg (HFD + FA) for 8 weeks.	The rats fed with HFD + FA had significantly lower plasma lipids and glucose levels compared with the HFD group.	[74]
	Daily dietary supplementation of male Wistar rats with Rosemary leaves (11–110 mg) rich in RA	Rosemary attenuated cardiac function improving metabolism & decreasing oxidative stress.	[79]
LIVER PROTECTION	Activity of CA on 1,3-dichloro-2-propanol-induced hepatotoxicity in rats that received CA (10 or 20 mg/kg bw) for 7 days.	CA protected against hepatotoxicity by enhancing the cytoprotective enzymes and lowering inflammation.	[85]
	Dietary supplementation of fish (*Megalobrama amblycephala*) with FA at 50–100 mg/kg bw	FA decreased pro-inflammatory cytokines alleviating acute liver injury.	[88]
	Rats exposed to aflatoxin B_1_AFB_1_ (75 µg/kg bw) were treated with GA (20 or 40 mg/kg bw) for 28 days.	GA ameliorated AFB_1_-induced hepatorenal dysfunction by decreasing oxidative stress and inflammation in rats hepatotoxicity.	[89]
	32 rats exposed to hepatic ischaemia/reperfusion injury were subsequently treated with RA dose of 50 mg/kg via oral gavage.	RA significantly reduced oxidative stress and abnormal histopathological findings in liver.	[92]
NEURO-PROTECTION	Systemic administration of neuroinflammatory rat with GA (100 mg/kg)	Clear neuroprotective effect of GA in treated rats compared to placebo	[106]
	Transgenic mice supplemented orally with epigallocatechin-3-gallate (EGCG) and/or FA (30 mg/kg each) daily for 3 months data	The combined EGCG-FA treatment reversed cognitive impairment, presenting AD therapeutic effect.	[104]
	Dietary supplementation of rats with 500 mg/kg body weight) of methanolic extracts of Salvia splendens (rich in RA and CA) for 4 weeks	The treatment significantly attenuated AlCl_3_-induced behavioral impairment (AD like).	[111]
	VA was tested against Fe^2+^- induced oxidative toxicity in brain tissues (neuronal cell lines—HT22).	VA exerted a clear neuroprotective activity.	[109]

CA: caffeic acid, FA: ferulic acid, GA: gallic acid, RA: rosmarinic acid, VA: vanillic acid, bw: body weight, AD: Alzheimer’s disease.

**Table 2 molecules-26-05405-t002:** Overview of the most recent studies on the potential activity of phenolic acids against microbial and viral relevant infections, including SARS-CoV-2.

Phenolic Acid Treatment	Activity against Microorganism/Infections	References
CA enhanced with chelated inorganic ions (or a metal such as iron)	Antiviral activity towards herpes simplex (HSV), VSV-Ebola pseudotypes and vaccinia viruses occurred early in the virus replication cycle.	[123]
CA (1.5 mg/mL) and chlorogenic acid (3 mg/mL)	Phenolic cocktail significantly inhibit the growth of the food born pathogen *Listeria monocytogenes*.	[99]
Methanol extracts of the leaf and stem of *E. africana* (rich in GA)	Antioxidant and antibacterial activity against *Staphylococcus* (*S. aureus* and *S. pyogenes*) with a minimum inhibitory concentration of 1.56 mg/mL—great potential for treatment of open wounds	[126]
CA and GA	CA and GA were found to exert good anti-viral activity against human coronavirus NL63 (HCoV-NL63)	[132]
CA and p-CA	Phenolic acids were found to bind to cell-surface HSPA5 competing for recognition by SARS-CoV-2 spike protein	[129]
Screening of 27 CA derivatives against 5 proteins of SARS-CoV-2	5 CA derivatives exerted anti-viral efficacy against COVID-19 via molecular docking and molecular dynamics simulation.	[133]

CA: caffeic acid, GA: gallic acid, *p*-CA: coumaric acid.

## Data Availability

Not applicable.

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
