# Peer review of "A Review of the Health Protective Effects of Phenolic Acids against a Range of Severe Pathologic Conditions (Including Coronavirus-Based Infections)"

_molecules, 2021, doi:10.3390/molecules26175405_

Round 1

Reviewer 1 Report

The review article entitled “A review on the health protective effects of phenolic acids against a range of severe pathologic conditions (incl. coronaviruses-based infections)” by Kiokias S et al., reports on the main structural features of phenolic acids, their most important natural sources and  their  extraction  techniques.

Studying phenolic acids from herbs, edible plants extracts is a research field of great interest, especially in the last few years. However, there some major  changes to address, as reported below:

Major changes:

Authors affirm that ‘‘A few authors have reviewed the radical scavenging capacity of phenolic acids and their subsequent beneficial effects against the development of cancer, cardiovascular diseases and  other  health  disorders  (such  as skin  problems,  inflammations,  bacterial  infections etc.)’’ In literature, there are many studies on these for example: N. Kaur et al., Nutrients 2021, 13, 2055;  A.-V. Pop (Cuceu) et al., 2015, Bulletin UASVM Food Science and Technology 72(2), 210 – 214, Please add a reference.

In section 3.2.2.  the authors claim that some medicinal plants have a high level of phenolic acids. This is right; however, they forget to mention that also some mountains food products such as mushrooms that are also a great source of phenolic compounds. See for example:

- Fogarasi M et al., Molecules, 2018, 23(9), 2261

Author Response

Thank you for your helpful and interesting comments Please see attached a separate document with the responses of the authors to your comments along  with the revised manuscript

Reviewer 2 Report

Phenolic acids are compounds commonly found in plants. They are divided into two groups: hydroxycinnamic acids and hydroxybenzoic acids. Currently, they are the object of increasing attention from nutritionists, researchers and experts in the sector, thanks to their antioxidant properties, capable of helping to keep cholesterol levels controlled, and in the prevention of some diseases. This review reports the main structural characteristics of phenolic acids, their most important natural sources and their extraction techniques. In addition, it provides an overview of the most recent clinical studies of their health effects.

In general

The topic is not particularly new, but it provides useful information and falls within the Journal's targets. The interesting aspect of the manuscript is the examination of phenolic acids from both a structural and an applicative point of view. In addition, it reports some studies on the effects of phenolic acids on infections, including coronavirus-based ones, spurring further research.

In detail

The review contains current scientific topics. The Introduction section adequately introduces the reader to the topic.

The structure of the text is adequate.

The references are up-to-date and reflect the state of international knowledge.

However, it is necessary to make some corrections/additions to the text.

Correct the typo, line 48: carnosic instead of carnocic.

Write in italics all plant names.

Numerous acronyms appear in the text, I suggest the authors to summarize them in a table to facilitate the reader.

Important: check the numbering of the references which does not correspond to the text.

Line 454: reference 132 is not present in the references section.

Line 806: insert reference 131 as indicated in the authors' guide.

Author Response

Thank you for your helpful and very interesting comments. In response, please see attached a document with the replies of the authors to your comments along with the revised manuscript 

Round 2

Reviewer 1 Report

The authors had improved sufficiently the quality and content of the manuscript by addressing the comments of the reviewer. In my opinion the paper is suitable for publication in the current form.